# Machine Learning Algorithm for Distinguishing Ductal Carcinoma In Situ from Invasive Breast Cancer

**DOI:** 10.3390/cancers14102437

**Published:** 2022-05-15

**Authors:** Vu Pham Thao Vy, Melissa Min-Szu Yao, Nguyen Quoc Khanh Le, Wing P. Chan

**Affiliations:** 1International Master Program of Medicine, Taipei Medical University, Taipei 110, Taiwan; m142109002@tmu.edu.tw; 2Department of Radiology, Thai Nguyen National Hospital, Thai Nguyen 24000, Vietnam; 3Department of Radiology, Wan Fang Hospital, Taipei Medical University, Taipei 110, Taiwan; wingchan@tmu.edu.tw; 4Department of Radiology, School of Medicine, College of Medicine, Taipei Medical University, Taipei 110, Taiwan; 5Professional Master Program in Artificial Intelligence in Medicine, College of Medicine, Taipei Medical University, Taipei 106, Taiwan; khanhlee@tmu.edu.tw; 6Research Center for Artificial Intelligence in Medicine, Taipei Medical University, Taipei 106, Taiwan

**Keywords:** ductal carcinoma in situ, minimally invasive breast cancer, XGBoost, mammographic, ultrasonographic, breast cancer

## Abstract

**Simple Summary:**

Breast cancer nowadays is the most common cancer among women. Two types refer to whether cancer has spread or not: Non-invasive and invasive breast cancers. Invasive ductal carcinoma is responsible for approximately 80% of all breast cancers, and ductal carcinoma in situ accounts for the majority of the remainder. Early identification of types of breast cancers provides breast cancer patients with more options for less invasive therapy. Our study aimed to develop a machine-learning classification model to differentiate ductal carcinoma in situ and minimally invasive breast cancer using clinical characteristics, mammography findings, ultrasound findings, and histopathology features. Our model showed that the five most important features were calcifications on mammograms, lymph node presence, microcalcifications on histopathology, the shape of the mass on ultrasound, and the orientation of the mass on ultrasound.

**Abstract:**

Purpose: Given that early identification of breast cancer type allows for less-invasive therapies, we aimed to develop a machine learning model to discriminate between ductal carcinoma in situ (DCIS) and minimally invasive breast cancer (MIBC). Methods: In this retrospective study, the health records of 420 women who underwent biopsies between 2010 and 2020 to confirm breast cancer were collected. A trained XGBoost algorithm was used to classify cancers as either DCIS or MIBC using clinical characteristics, mammographic findings, ultrasonographic findings, and histopathological features. Its performance was measured against other methods using area under the receiver operating characteristic curve (AUC), sensitivity, specificity, accuracy, precision, and F1 score. Results: The model was trained using 357 women and tested using 63 women with an overall 420 patients (mean [standard deviation] age, 57.1 [12.0] years). The model performed well when feature importance was determined, reaching an accuracy of 0.84 (95% confidence interval [CI], 0.76–0.91), an AUC of 0.93 (95% CI, 0.87–0.95), a specificity of 0.75 (95% CI, 0.67–0.83), and a sensitivity of 0.91 (95% CI, 0.76–0.94). Conclusion: The XGBoost model, combining clinical, mammographic, ultrasonographic, and histopathologic findings, can be used to discriminate DCIS from MIBC with an accuracy equivalent to that of experienced radiologists, thereby giving patients the widest range of therapeutic options.

## 1. Introduction

Among women, breast cancer is the most common cancer in the world aside from nonmelanoma skin cancers [1]. According to the World Health Organization, 2.3 million women were diagnosed with breast cancer in 2020, leading to 685,000 deaths worldwide. Breast cancers are divided into two types, non-invasive and invasive, based on whether it has spread. Among the non-invasive types, ductal carcinoma in situ (DCIS) is the most common [2], accounting for around 84% of all in situ cancers [3]. Pathologists further divide DCIS into four forms: papillary, cribriform, solid, and comedo [4]. High-grade DCIS (comedo) is the quickest to progress to an invasive form. The typical treatment for DCIS has been mastectomy; however, breast conservation therapies for invasive breast cancers have gained acceptance, and initial attempts at using breast-conserving surgeries for DCIS are potentially acceptable [3].

Breast cancers become invasive when they grow outside of the ducts or lobules into the adjacent breast tissue. Up to 70% of these cases have been identified as invasive ductal cancer, also known as infiltrating ductal carcinoma [5], wherein smaller tumor size is associated with a greater survival rate. According to Tabár et al. [6], women with invasive tumors no larger than 14 mm in size and accompanied by casting-type calcifications had a 20-year survival rate of 55%. Mammography screening and improved therapies have increased long-term survival prognoses for patients with invasive carcinomas in this size range [7]. With early identification of cancer type, a greater range of options for those seeking less-invasive therapies becomes available. Many studies have shown the importance of traditional histopathological characteristics in the prediction of breast cancer, such as lymph node status, tumor size, histological grade, margin width, and several biological indicators [8,9,10]. These prognostic factors are appealing in principle and effective with large tumors, but they present challenges when tumors are small [11].

Machine learning (ML) is a part of artificial intelligence that focuses on developing computer algorithms that can change when new information is added. Several studies have shown that ML techniques can be used to quickly diagnose breast cancer with great accuracy [12,13]. However, to our knowledge, no studies have shown that ML algorithms can be used to discriminate between DCIS and minimally invasive breast cancer (MIBC). This study was therefore designed to develop an ML classification model to differentiate DCIS from MIBC using clinical characteristics, mammographic and ultrasonographic findings, and histopathologic features. A successful model can support radiologists in disease diagnosis and decrease the time required to do so.

## 2. Materials and Methods

The Taipei Medical University Joint Institution Review Board approved this study (TMU-JIRB No. N202203003), and patient informed consent was waived due to its retrospective nature.

### 2.1. Study Design

We defined DCIS as discrete spaces filled with malignant cells, frequently surrounded by a recognizable basal cell layer containing normal myoepithelial cells [4]. We defined MIBC as invasive breast cancer found to be less than or equal to 15 mm in size when assessed histologically. Patients who underwent biopsies to confirm breast cancer between 1 January 2010 and 31 December 2020 at Wanfang Hospital of Taipei Medical University were considered eligible for this study (*n* = 1377), leading to electronic medical records reviews. Those lacking tumor measurements in pathology or with insufficient histopathologic information (*n* = 68) were excluded, leaving 1309 patients to be consecutively enrolled in the study. Of these, 245 were found to have DCIS, and 1064 were found to have invasive breast cancer. From the DCIS group, 56 patients were excluded due to microinvasion, as shown via pathology. From the MIBC group, 833 patients were excluded due to excess tumor size (>15 mm). Finally, 420 women with either pure DCIS (*n* = 189) or MIBC (*n* = 231) were included in the study (Figure 1).

### 2.2. Data Acquisition

Our final patient set was split into three non-overlapping sets: 70% (294 patients) for training, 15% (63) for validation, and 15% (63) for testing. Medical records were reviewed to retrieve clinical data, such as age, body mass index (BMI), menopausal status, age at menarche, age at first live birth, family history of breast cancer, use of hormone replacement therapy, and clinical signs of cancer (palpable vs not palpable).

Sonographic features were interpreted using BI-RADS criteria (5th ed.) [14], specifically, breast composition and the breast mass features tumor size, shape, orientation, margin, echo pattern, posterior features, calcifications, vascularity, and elasticity assessment. Architectural distortion, ductal changes, and the status of the axillary lymph nodes were noted. Interval changes on follow-up ultrasound (US) examinations were recorded as well. The sonographic features used in the models are provided in detail in Appendix A (Table A1).

Mammographic findings were also interpreted using BI-RADS criteria [14], specifically those of the masses (size, shape, density, and margin), calcifications (morphology and distribution), and architectural distortion. Asymmetries in density and morphology were also recorded. Interval changes on follow-up mammograms (MMGs) were also reviewed. The MMG features used in the models are provided in detail in Appendix A (Table A1).

Histopathologic findings from excisional biopsies or mastectomy specimens were used as gold standards. The histologic parameters recorded were nuclear grade, presence of comedo necrosis, architectural pattern, and the expressions of ER, PR, and HER-2 (Table A1).

### 2.3. Model Development

The process used to develop the classification model is shown in Figure 2. Preprocessing improves the quality of a dataset, supplying clean data that can be used for modeling [15]. In this study, we processed missing values, selected correlation-based features, and labelled features using One Hot Encoder.

Data selection allows the fittest features to be chosen after ranking them using a training dataset. Feature selection, on the other hand, is choosing the combination of features important for classification in preference to those that are less important. The feature selection techniques used in this study were recursive feature elimination methods and application of the XGBoost [16] ‘Feature Importance Scores,’ applying SHapley Additive exPlanations (SHAP) [17] methods.

To develop our model, we used a gradient boosting-based decision-tree-based ensemble ML algorithm in XGBoost in which the computational complexity of determining the optimal split, typically the most time-consuming element of decision tree building methods, is reduced. To find the most important features, multiple values for k were tested using the Select K Best algorithm. We trained the model based on selected features’ importance from the training dataset and wrapped it in the SelecFromModel algorithm. After fitting it with input data, this algorithm extracts the most viable features based on the importance of model weights. Then, we selected the features to be used with the testing dataset to evaluate the model.

Using the same set of features, our model’s performance was compared against other ML algorithms, such as random forest, single vector machine, Gaussian naive Bayes, K-nearest neighbor, and decision tree classifier. The hyperparameters of the XGBoost model were manually tuned and fixed throughout the training process by comparing errors during training, validating using the testing dataset, and automating the determination of the best-fit hyperparameters using a grid search method. To generate more robust models and avoid overfitting, k-fold cross validation was applied when using XGBoost. The hyperparameters of our model are provided in the Table A2 (Appendix B).

We also compared model performance (i.e., classifying DCIS vs MIBC) with the diagnostic performance of radiologists. The testing set of patients was separated into three groups: those to be used with MMG alone, those to be used with US alone, and those where both were used. We also compared the sensitivities and specificities of the groups that comprised the entire sample. Each was independently assessed by 2 radiologists of Wanfang hospital (Radiologist 1 was a first-year resident, and Radiologist 2 had more than ten years of breast imaging experience) for diagnoses. All diagnoses by radiologists and by the model were compared with pathological results. Model performance was evaluated using a number of metrics: accuracy, area under the receiver operating characteristic curve (AUC), precision, recall, sensitivity, and specificity.

### 2.4. Statistical Analysis

Statistical analyses were performed using SPSS, version 25 (SPSS Inc., Chicago, IL, USA). Clinical characteristics of the two types of cancers were compared using the χ^2^ test for categorical variables and the Mann–Whitney test for continuous variables. The McNemar test for sensitivity and specificity was used to compare the diagnostic performance of the model with those of the radiologists. The significance of the differences between evaluation metrics was estimated using the 95% confidence interval (CI), using *p* < 0.05 to find significant differences.

The evaluation metrics were calculated as follows:Precision = $\frac{\mathrm{TP}}{\mathrm{TP}+\mathrm{FP}}$Recall = $\frac{\mathrm{TP}}{\mathrm{TP}+\mathrm{FN}}$Accuracy = $\frac{\mathrm{TP}+\mathrm{TN}}{\mathrm{TP}+\mathrm{FP}+\mathrm{TN}+\mathrm{FN}}$F1 score = $\frac{2\;x\;\mathrm{Precision}\;x\;\mathrm{Recall}}{\mathrm{Precision}+\mathrm{Recall}}$Specificity = $\frac{\mathrm{TN}}{\mathrm{TN}+\mathrm{FP}}$Sensitivity = $\frac{\mathrm{TP}}{\mathrm{TP}+\mathrm{FN}}$
where TP, FP, TN, and FN are true positive, false positive, true negative, and false negative, respectively.

## 3. Results

### 3.1. Study Population

The clinical characteristics of our study group, both as a whole and as divided by cancer type, are summarized in Table 1. The characteristics of the two groups are similar, showing statistically significant differences only in age at first live birth, family history of breast cancer, and BMI group (*p* < 0.05). A larger portion of the DCIS group bore their first child between the ages of 20 and 29 years compared to the MIBC group. Conversely, a larger portion of the MIBC group were nulliparous. The DCIS group had a greater frequency of familial breast cancer history (*p* < 0.05) and a lower frequency of low BMIs (<18.5 kg/m^2^). The MIBC group had the greatest frequency of BMIs in the 24–27 kg/m^2^ range (*p* < 0.05).

As shown in Table 2, the training set consisted of 294 patients (mean [standard deviation] age, 56.8 [11.5] years; BMI, 24.0 [4.6] kg/m^2^) of which 130 (44%) were diagnosed with DCIS. On the other hand, the testing set contained 35 (55.5%) patients with DCIS.

### 3.2. Model Development

#### 3.2.1. Missing Value Processing

We examined 187 features across sonographic, mammographic, and histopathologic findings. Of the 420 patients in our study group, MMG and US were not performed in 99 (24%) and 22 (5%) of the patients, respectively, leading to a number of missing features in those patients. The degree to which these initial features were missing is shown in Figure 3. Those features that were missing more than 30% of the time were excluded from the model. Those features that were missing less than 30% of the time were imputed when missing.

#### 3.2.2. Correlation-Based Feature Selection

After imputing the missing values, a correlation analysis of the features was performed to avoid using features that were highly correlated with the other features, creating a linear correlation and having little additional impact on the dependent variable. When the correlation exceeded 0.8, that feature was eliminated from the dataset in favor of those with lower means. The correlation analysis is visualized in Figure 4.

### 3.3. Performance of XGBoost and Feature Importance Analysis

Using the final feature set, the final data were entered into XGBoost, yielding a model accuracy of 0.79 (95% CI, 0.72–0.83) and an AUC of 0.81 (95% CI, 0.73–0.84) as a baseline. To improve on that, a feature importance analysis was used to rank feature importance based on their scores as determined by the XGBoost classifier combined with Select K Best. A set of 147 features (k = 147) yielded the greatest accuracy, F1 score, recall, and precision. As shown in Figure 5, The AUC of the testing dataset reached 0.93 for the breast cancer classification task, producing an overall accuracy of 0.84. Model sensitivity was 0.91 (95% CI, 0.76–0.94) and specificity was 0.75 (95% CI, 0.67–0.83). The scores of the 147 features, based on the XGBoost model, are shown in Figure 6.

By using the SHAP method, we found the 20 most important features that had the most influence on positive prediction of MIBC (Figure 6). See Appendix B (Table A3 and Table A4) for the feature contribution analysis corresponding to this figure.

### 3.4. Performance as Compared with Other ML Methods

The performance of this classification model, using XGBoost, was compared with those of four other algorithms, using the F1 score, recall score, accuracy, precision, and AUC. Results are shown in Table 3 and Figure 5. All five methods showed good results in F1 score, accuracy, and recall; however, XGBoost and the random forest classifier models performed the best using these metrics.

### 3.5. Performance as Compared with Radiologists

Classification by radiologist, when performed using MMG alone or using US alone, yielded lower sensitivity and specificity compared to the use of MMG plus US are shown in Table 4. Compared to the model, Radiologist 1 achieved significantly lower performance metrics for classifying DCIS vs MIBC (*p* < 0.05), achieving a specificity of 0.64 (95% CI, 0.57–0.66) and a sensitivity of 0.74 (95% CI, 0.68–0.79). On the other hand, Radiologist 2 achieved sensitivity and specificity metrics similar to those of the model (*p* > 0.05).

## 4. Discussion

The ML model used clinical features, mammographic features, ultrasonographic features, and histopathologic features extracted from patient medical records to classify DCIS and MIBC. It achieved an AUC of 0.93 (95% CI, 0.87–0.95), a sensitivity of 0.91, and a specificity of 0.75.

Our results support those of others who showed that the age at which a woman bears her first child, a family history of breast cancer, and BMI were associated with the ability to distinguish DCIS from MIBC. Louise et al. reported significant trends in identifying invasive cancer based on the age at which a woman’s first child is born, showing relative risks on the order of 2.2 to 2.7 when the first child is born after 30 years of age compared to before 20 years of age [15]. They further found that a woman’s BMI was slightly associated with the risk of small invasive lesions [15]. The key contributors for classifying DCIS vs. MIBC, therefore, might be age at first live birth, family history of breast cancer, and BMI.

The ML model developed in this study used clinical features, MMG features, US features, and histopathological features to distinguish DCIS from MIBC. When interpreting US and MMG features, BI-RADS criteria (5th ed.) were used. Using all the features, the XGBoost model achieved an AUC of 0.81 (95% CI, 0.73–0.83). By ranking the features and selecting a subset of 147 features based on k score, the model was trained. The five features that contributed the most to the classification task were identified by SHAP analysis as: the appearance of calcification on the MMG, a non-parallel orientation of the mass on US imaging, the presence of microcalcification as identified by histopathology, enhancement of the posterior feature of the mass on US imaging, and BMI group. On the other hand, XGBoost identified the five most important features as: the appearance of calcification on the MMG, the existence of lymph nodes, the presence of microcalcification as identified by histopathology, an irregular shape of the mass on US imaging, and a non-parallel orientation of mass on US imaging. These findings are consistent with those of others. Compared to DCIS, invasive tumors are more irregular in shape, non-parallel in orientation, and yield a hypoechoic or complicated echo pattern [18]. Chen et al. [19] reported that an internal echo pattern was the most important feature differentiating invasive cancers from DCIS.

When four additional ML methods were used, the best-performing model was the random forest classifier, but XGBoost achieved better performance metrics across the board. Furthermore, a multilayer cross-validation method was used to optimize model hyperparameters and avoid overfitting, thereby boosting model generalizability. Several studies have applied artificial intelligence on oriented radiological tasks, particularly those that differentiate breast cancers as benign or malignant. For example, in 2015, Mandeep Rana et al. [20] developed a Support Vector Machines sequential minimum optimization model that combined a K-nearest neighbors algorithm approach with Manhattan measures and other ML techniques to classify breast cancers. In 2018, Maysanjaya et al. [21] combined two algorithms to develop a Computer-aided Detection based method and a naive Bayes algorithm that achieved an accuracy of 99.27%. Ezgi Mercan et al. [22] built a classification model to discriminate between invasive and non-invasive breast cancer based on breast pathology structures using Digital WSIs for breast biopsies. The accuracy of the model reached 0.98, and the sensitivity was 0.84. In addition, Shikha Roy et al. [23] demonstrated that DCIS and invasive ductal carcinoma can be classified based on gene expression with RNA-seq gene expression profiles from The Cancer Genome Atlas (TCGA). In addition, Niyazi Senturk et al. [24] proposed an AI model to assess the risk of BRCA variation of breast cancer. Unlike the models in these studies, which required the use of many algorithms and a large number of images, ours uses only XGBoost with optimization, reducing training time and model complexity. Furthermore, we focused on distinguishing DCIS from MIBC rather than benign from malignant, a greater challenge for radiologists.

Our model could classify breast cancers with a specificity and sensitivity similar to or greater than those achieved by our radiologists. Greater sensitivities and specificities were achieved by the radiologists when using both MMG and US images compared to using only one of these imaging modalities. This confirms the results of others who have also shown better breast cancer detection when using both MMG and US imaging [25,26]. Compared to a first-year resident, our model’s specificity and sensitivity were greater (0.75 vs. 0.64, *p* < 0.05; and 0.91 vs. 0.74, *p* < 0.05, respectively). Compared to a ten-year veteran radiologist, even with using MMG plus US image, however, the differences were not significant. Even though the ML model cannot replace diagnostics by radiologists, their workloads can be reduced when this ML model is implemented.

This study has several limitations. First, it was implemented in a single center, and external validation of the best-performing model was not performed. Doing so could have further demonstrated its generalizability. Second, it was a retrospective study based on a limited number of patients. Therefore, studies employing larger sample sizes are needed to confirm our results. On the other hand, a deep learning approach, such as a convolutional neural network, might outperform this model when combined with radiomic features (such as MMG image or magnetic resonance images) or genetic data to enhance the performance of our model.

## 5. Conclusions

In conclusion, the XGBoost model developed in this study, when provided with clinical characteristics, mammographic and ultrasonographic findings, and histopathologic features from medical records, can successfully discriminate DCIS from MIBC at the level of experienced radiologists, thereby providing patients with more options for less-invasive therapies.

## Figures and Tables

**Figure 1 cancers-14-02437-f001:**
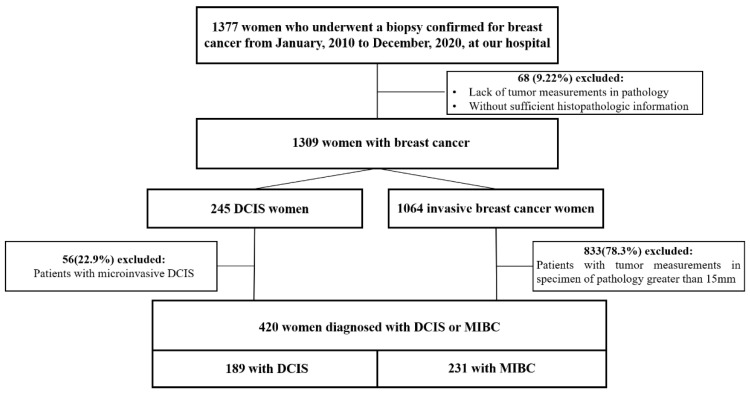
Flowchart of the study population. DCIS: ductal carcinoma in situ; MIBC: minimally invasive breast cancer.

**Figure 2 cancers-14-02437-f002:**
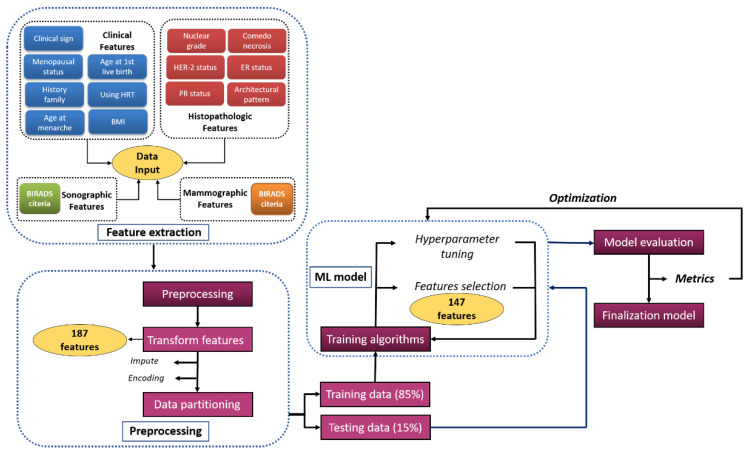
Workflow for the machine learning (ML) model used to distinguish ductal carcinoma in situ from minimally invasive breast cancer.

**Figure 3 cancers-14-02437-f003:**
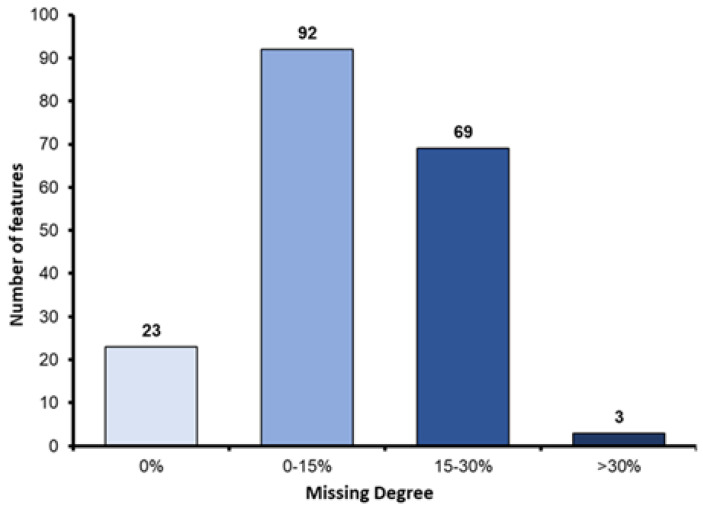
Initial degree of missing features.

**Figure 4 cancers-14-02437-f004:**
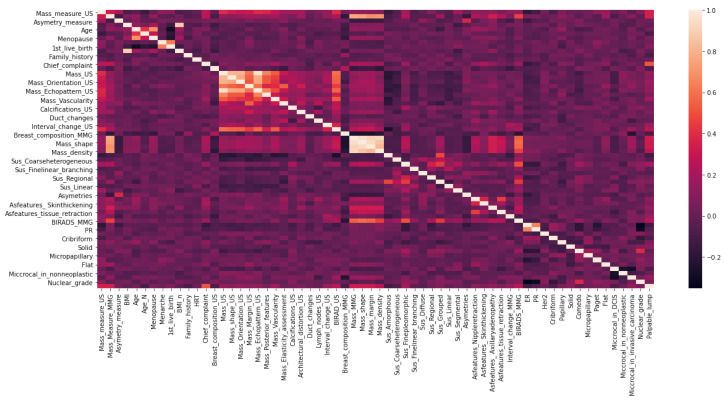
Correlation analysis.

**Figure 5 cancers-14-02437-f005:**
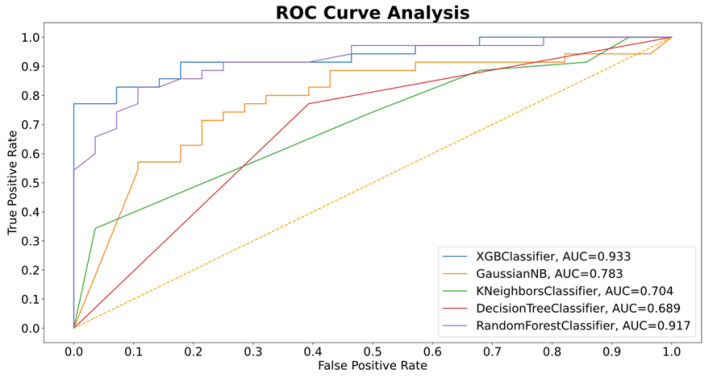
Performances of five models based on area under the receiver operating characteristic curve (AUC).

**Figure 6 cancers-14-02437-f006:**
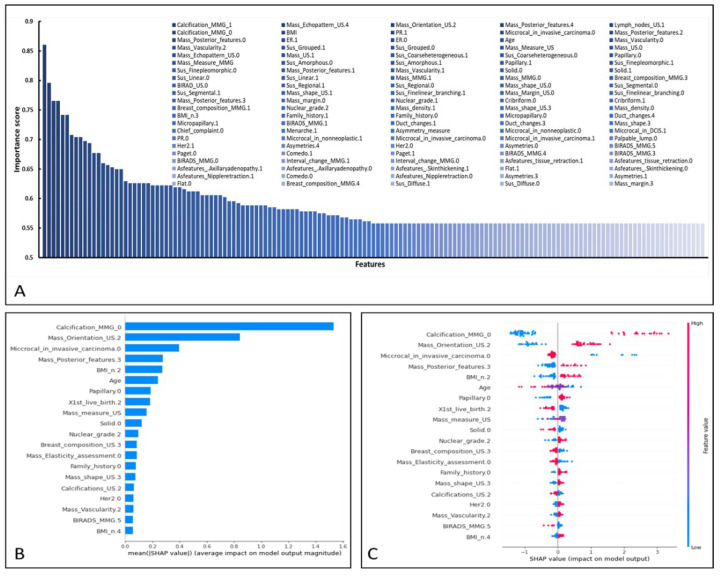
(**A**) Feature importance according to XGBoost. (**B**,**C**) Contribution of the top 20 features as ranked by SHapley Additive exPlanations (SHAP). The features are arranged in descending order on the *y*-axis according to their mean absolute influence on classification. Each dot represents the SHAP value for a certain feature for a certain patient. The SHAP algorithm evaluates all conceivable combinations of features, including and excluding a given feature to evaluate its contribution to the prediction. The farther away from the *y*-axis (positive or negative *x*) a dot is placed, the more impact this attribute has on the machine learning model output for that woman. Dot color indicates the feature’s original value from low (blue) to high (magenta), as indicated by the color array stripe on the right. The color was determined separately for each feature based on the patient’s feature values.

**Table 1 cancers-14-02437-t001:** Clinical characteristics of the study population.

Characteristic	Study Group (*n* = 420)	DCIS Group (*n* = 189)	MIBC Group (*n* = 231)	*p* Value ^b^
Age ^a^, y	57.1 (12.0)	57.1 (12.0)	57.3 (12.0)	0.694
Age group				0.086
<40 y	22 (5.2)	6 (3.2)	16 (6.8)	…
≥40 y	398 (94.8)	183 (96.8)	215 (93.2)	…
Menopause				0.643
Premenopause	145 (34.5)	63 (33.3)	82 (35.3)	…
Postmenopause	275 (65.5)	126 (66.7)	149 (64.5)	…
Age at menarche				0.837
NA	93 (22.1)	44 (23.2)	49 (21.2)	…
<12 y	27 (6.4)	13 (6.9)	14 (6.1)	…
12–14 y	222 (52.9)	99 (52.4)	123 (53.2)	…
≥15 y	78 (18.6)	33 (17.5)	45 (19.5)	…
Age at first live birth				0.002
<20 y	12 (3.6)	1 (0.7)	11 (6.0)	…
20–29 y	166 (49.7)	91 (60.3)	75 (41.0)	…
≥30 y	82 (24.6)	34 (22.5)	48 (26.2)	…
Nulliparous	74 (22.1)	25 (16.5)	49 (26.8)	…
Family history of BC				0.002
Yes	83 (19.8)	50 (26.5)	33 (14.3)	…
No	333 (80.2)	139 (73.5)	198 (85.7)	…
History of HRT use				0.464
Yes	31 (7.4)	12 (6.3)	19 (8.2)	…
No	389 (92.6)	177 (93.7)	212 (91.8)	…
BMI ^a^ (kg/m²)	24.02 (4.40)	24.02 (4.39)	24.00 (4.41)	0.542
BMI group				0.002
BMI < 18.5 kg/m²	14 (3.33)	4 (2.12)	10 (4.32)	…
18.5 ≤ BMI < 24 kg/m²	227 (54.05)	98 (51.85)	129 (55.84)	…
24 ≤ BMI < 27 kg/m²	82 (19.52)	44 (23.28)	38 (16.45)	…
BMI ≥ 27 kg/m²	97 (23.10)	43 (22.75)	54 (23.39)	…

Unless otherwise indicated, data in the table are expressed as number (percentage) ^a^ Expressed as mean (standard deviation)**.**
^b^ DCIS group vs. MIBC group. DCIS: ductal carcinoma in situ; MIBC: minimally invasive breast cancer; NA: not available; HRT: hormone replacement therapy; BMI: body mass index; BC: breast cancer.

**Table 2 cancers-14-02437-t002:** Characteristics of the training and testing sets.

Characteristic	Training Set	Testing Set	*p* Value
No. of patients	357	63	
DCIS	161 (45.1)	35 (55.5)	
MIBC	196 (54.9)	28 (45.4)	
Age ^a^, y	57.1 (11.6)	58.5 (12.8)	>0.05
BMI ^a^, kg/m^2^	24.1 (4.7)	24.1 (4.9)	>0.05
Premenopause	124 (34.7)	21 (33.3)	>0.05
Postmenopause	233 (65.3)	42 (66.7)	>0.05
Family history of BC	61 (18.7)	16 (25.4)	>0.05

Unless otherwise indicated, data in the table are expressed as number (percentage) ^a^ Expressed as mean (standard deviation). DCIS: ductal carcinoma in situ; MIBC: minimally invasive breast cancer; BMI: body mass index; BC: breast cancer.

**Table 3 cancers-14-02437-t003:** Performance comparisons of five models.

Model	Accuracy	F1 Score	Recall	Precision
XGBoost	0.84 [0.76–0.91]	0.87 [0.79–0.93]	0.91 [0.76–0.94]	0.82 [0.71–0.92]
GaussianNB	0.75 [0.67–0.84]	0.79 [0.67–0.86]	0.88 [0.68–0.93]	0.72 [0.65–0.92]
KNeighbors Classifier	0.63 [0.54–0.69]	0.73 [0.56–0.80]	0.87 [0.57–0.92]	0.62 [0.55–0.90]
DecisionTree Classifier	0.73 [0.64–0.82]	0.76 [0.64–0.84]	0.77 [0.64–0.86]	0.75 [0.64–0.86]
RandomForest Classifier	0.82 [0.74–0.89]	0.84 [0.76–0.91]	0.89 [0.73–0.93]	0.81 [0.78–0.91]

Data in the table are expressed as value [95% confidence interval].

**Table 4 cancers-14-02437-t004:** Performance comparison of the XGBoost model and two radiologists.

	Sensitivity	*p^se^*	Specificity	*p^sp^*
Radiologist 1				
Using MMG alone	0.65 (0.61–0.71)		0.59 (0.57–0.62)	
Using US alone	0.67 (0.62–0.72)		0.59 (0.55–0.63)	
Using both US and MMG	0.74 (0.68–0.79)	<0.05	0.64 (0.57–0.66)	<0.05
Radiologist 2				
Using MMG alone	0.81 (0.74–0.86)		0.68 (0.65–0.72)	
Using US alone	0.77 (0.73–0.82)		0.64(0.61–0.74)	
Using both US and MMG	0.83 (0.74–0.88)	>0.05	0.71 (0.68–0.74)	>0.05
XGBoost Model	0.91 (0.76–0.94)		0.75 (0.68–0.78)	

Data in the table are expressed as value (95% confidence interval)**.** DCIS: ductal carcinoma in situ; MIBC: minimally invasive breast cancer; MMG: mammogram; US: ultrasound. *p^se^*, *p^sp^* indicate the probability of significant differences in sensitivity and specificity, respectively, between the XGBoost model and the radiologist.

## Data Availability

The data that support the findings of this study are available from the corresponding author upon reasonable request.

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
