# Peer review of "Machine Learning Algorithm for Distinguishing Ductal Carcinoma In Situ from Invasive Breast Cancer"

_cancers, 2022, doi:10.3390/cancers14102437_

Round 1
Reviewer 1 Report
The authors aimed to develop a machine-learning classification model to differentiate ductal carcinoma in situ and minimally invasive breast cancer using clinical characteristics, mammography findings, ultrasound findings, and histopathology features. The authors showed that the five most important features were calcifications on mammograms, lymph node presence, microcalcifications on histopathology, the shape of the
mass on ultrasound, and the orientation of the mass on ultrasound.
Authors mentioned also study limitations and well discussed these limitations.
The authors concluded that the XGBoost model developed in this study, when provided with clinical characteristics, mammographic and ultrasonographic findings, and histopathologic features from medical records, can successfully discriminate DCIS from MIBC at the level of experienced radiologists, thereby providing patients with more options for less insive therapies.
Genetic data would be integrated this AI beased decision support system and at least it can be discussed as future directions.
The study is unique but should be improved.
There are several updated studies using AI in breast cancer. The authors did not use and mention these studies such as :
JAMA Network Open JAMA Netw Open. 2019 Aug; 2(8): e198777. Published online 2019 Aug 9. doi: 10.1001/jamanetworkopen.2019.8777 Scientific Reports2020 Mar 5;10(1):4113. doi: 10.1038/s41598-020-60740-w. Genes . 2021 Nov 9;12(11):1774. doi: 10.3390/genes12111774. References should be improved.Author Response
Response to Reviewer 1 Comments
General Comment: The authors concluded that the XGBoost model developed in this study, when provided with clinical characteristics, mammographic and ultrasonographic findings, and histopathologic features from medical records, can successfully discriminate DCIS from MIBC at the level of experienced radiologists, thereby providing patients with more options for less invasive therapies. Genetic data would be integrated into this AI-based decision support system and at least it can be discussed in future directions. The study is unique but should be improved. There are several updated studies using AI in breast cancer. The authors did not use and mention these studies.
General Response: Thank you for your insightful comments on our manuscript. As per your suggestion, we will use radiomic features or genetic data in the future study. Specifically, we have added updated references in the text accordingly.
Major Concerns:
(1) There are several updated studies using AI in breast cancer. The authors did not use and mention these studies such as JAMA Network Open JAMA Netw Open. 2019 Aug; 2(8): e198777. Published online 2019 Aug 9. doi: 10.1001/jamanetworkopen.2019.8777 Scientific Reports2020 Mar 5;10(1):4113. doi: 10.1038/s41598-020-60740-w. Genes. 2021 Nov 9;12(11):1774. doi: 10.3390/genes12111774. References should be improved.
Author's response: We have added updated studies using AI in breast cancer accordingly (page 11, 2nd paragraph).
Line 305-312: Ezgi Mercan et al. [20] built a classification model to discriminate between invasive and non-invasive breast cancer based on breast pathology structures using Digital WSIs for breast biopsies. The accuracy of the model reached 0.98 and the sensitivity was 0.84. In addition, Shikha Roy et al. [21] demonstrated that DCIS and invasive ductal carcinoma could be classified based on gene expression by RNA-seq gene expression profiles from The Cancer Genome Atlas (TCGA). In addition, Niyazi Senturk et al. [22] proposed an AI model to assess the risk of BRCA variation of breast cancer.
References:
- Mercan, E.; Mehta, S.; Bartlett, J.; Shapiro, L.G.; Weaver, D.L.; Elmore, J.G. Assessment of Machine Learning of Breast Pathology Structures for Automated Differentiation of Breast Cancer and High-Risk Proliferative Lesions. JAMA Network Open 2019, 2, e198777-e198777, doi:10.1001/jamanetworkopen.2019.8777
- Roy, S.; Kumar, R.; Mittal, V.; Gupta, D. Classification models for Invasive Ductal Carcinoma Progression, based on gene expression data-trained supervised machine learning. Scientific Reports 2020, 10, 4113, doi:10.1038/s41598-020-60740-w.
- Senturk, N.; Tuncel, G.; Dogan, B.; Aliyeva, L.; Dundar, M.S.; Ozemri Sag, S.; Mocan, G.; Temel, S.G.; Dundar, M.; Ergoren, M.C. BRCA Variations Risk Assessment in Breast Cancers Using Different Artificial Intelligence Models. Genes 2021, 12, doi:10.3390/genes12111774.
(2) Genetic data would be integrated into this AI-based decision support system and at least it can be discussed in future directions.
Author's Response: Thank you! We have revised accordingly.
Line 331-334: On the other hand, a deep learning approach, such as a convolutional neural network, might outperform this model when combined with radiomic features (such as MMG image or magnetic resonance images) or genetic data to enhance the performance of our model.

Reviewer 2 Report
Each of the findigns has been published previously in many different journals but the merit of the MS is by combining multiple factors in one algorythm.
I think that it is valauble MS
The MS did not cinlcude all previous refernces in all different parameters addressed previously. Recommend to add and acknowlege more pertinent references which have been addressed previously.
Author Response
Response to Reviewer 2 Comments
General Comment: Each of the findings has been published in several journals but the spotlight is this MS combined many factors in one algorithm. The MS did not include all previous references in all different parameters addressed previously. Recommend adding and acknowledging more relevant sources that have been addressed previously.
General Response: Thank you for your comments! The authors would like to combine many features from 4 groups: clinical, mammographic, ultrasonographic, and histopathological features to build a model that can classify Ductal carcinoma in situ and Minimally invasive breast cancer. As per your suggestions, we have added updated references in the text accordingly.
Major Concerns:
The MS did not include all previous references in all different parameters addressed previously. Recommend adding and acknowledging more pertinent references which have been addressed previously.
Author's Response: Thank you! Firstly, some parameters we found in this study are consistent with previous studies and we mention them in Discussion, lines 290-294.
Line 305-312: Ezgi Mercan et al. [20] built a classification model to discriminate between invasive and non-invasive breast cancer based on breast pathology structures by using Digital WSIs for breast biopsies. The accuracy of the model reached 0.98 and the sensitivity was 0.84. In addition, Shikha Roy et al. [21] demonstrated that DCIS and invasive ductal carcinoma could be classified based on gene expression by RNA-seq gene expression profiles from The Cancer Genome Atlas (TCGA). In addition, Niyazi Senturk et al. [22] proposed an AI models to assess risk from BRCA variation of breast cancer.
References:
- Mercan, E.; Mehta, S.; Bartlett, J.; Shapiro, L.G.; Weaver, D.L.; Elmore, J.G. Assessment of Machine Learning of Breast Pathology Structures for Automated Differentiation of Breast Cancer and High-Risk Proliferative Lesions. JAMA Network Open 2019, 2, e198777-e198777, doi:10.1001/jamanetworkopen.2019.8777
- Roy, S.; Kumar, R.; Mittal, V.; Gupta, D. Classification models for Invasive Ductal Carcinoma Progression, based on gene expression data-trained supervised machine learning. Scientific Reports 2020, 10, 4113, doi:10.1038/s41598-020-60740-w.
- Senturk, N.; Tuncel, G.; Dogan, B.; Aliyeva, L.; Dundar, M.S.; Ozemri Sag, S.; Mocan, G.; Temel, S.G.; Dundar, M.; Ergoren, M.C. BRCA Variations Risk Assessment in Breast Cancers Using Different Artificial Intelligence Models. Genes 2021, 12, doi:10.3390/genes12111774.

Reviewer 3 Report
I am really grateful for reviewing this manuscript. In my opinion, this manuscript can be published once some revision is done successfully. This study used a small sample (420) but achieved AUCs with the confidence interval of 87-95. I would like to point out that this is a great achievement. Also, this study combined demographic, mammographic, ultra-sonographic and histopathologic features, which I strongly argue is a rare breakthrough. But I would like to make one suggestion regarding the directions of SHAP values in Figure 6. The rationale for estimating the SHAP values of a particular predictor is to identify the direction of association between the dependent variable (DCIS vs. MIBC) and the predictor (e.g., Calcification_MMG_0). In other words, the SHAP values of a particular predictor are useful when these values have one direction, i.e., either negative or positive. Here, the SHAP value of a particular predictor for a particular participant measures a difference between what the XGBoost model predicts for the probability of DCIS vs. MIBC for the participant with and without the predictor. However, the SHAP values of every predictor are found to be spread between negative and positive extremes in Figure 6. I would like to suggest the authors to address this issue in detail.
Author Response
Response to Reviewer 3 Comments
General Comment: This manuscript can be published once some revision is done successfully. This study used a small sample (420) but achieved AUCs with a confidence interval of 87-95. This is a great achievement. Also, this study combined demographic, mammographic, ultra-sonographic, and histopathologic features. One suggestion regarding the directions of SHAP values in Figure. The SHAP value of a particular predictor for a particular participant measures a difference between what the XGBoost model predicts for the probability of DCIS vs. MIBC for the participant with and without the predictor. However, the SHAP values of every predictor are found to be spread between negative and positive extremes in Figure 6.
Author’s Response: Thank you for your insightful comment! In this study, we used the SHAP method to find the 20 most important features that had the most influence on the positive prediction of MIBC. As we mentioned in the legends of Figure 6, the x-axis stands for SHAP value, and the y-axis has 20 features. Each point on the chart is one SHAP value for a prediction and feature. The red color represents a higher value of a feature. Blue indicates the lower value of a feature. The SHAP algorithm analyzes all conceivable combinations of features, including and excluding a given feature to evaluate its contribution to the prediction. The farther away from the y-axis (positive or negative x) a dot is placed, the more impact this attribute has on the machine learning model output for that woman. Dot color indicates the feature's original value from low (blue) to high (magenta), as indicated by the color array stripe on the right. A positive SHAP value represents a positive impact on prediction, leading the model to predict 1 (MIBC). A negative SHAP value indicates a negative impact, leading the model to predict 0 (DCIS). For example, in this chart, we can conclude the following insights: A higher value of “Calcification_MMG_0” (Do not have suspicious morphology calcification on mammography) leads to MIBC and a lower value of “Calcification_MMG_0” ( Have suspicious morphology calcification on mammography) leads to DCIS. These findings are consistent with previous studies, Holland et al. [1] reported that microcalcifications are associated with ductal carcinoma in situ. Grimm et al. [2] demonstrated that most DCIS lesions are first diagnosed as calcifications on screening mammograms.
References:
- Holland, R.; Hendriks, J.H. Microcalcifications associated with ductal carcinoma in situ: mammographic-pathologic correlation. Seminars in diagnostic pathology 1994, 11, 181-192.
- Grimm, L.J.; Miller, M.M.; Thomas, S.M.; Liu, Y.; Lo, J.Y.; Hwang, E.S.; Hyslop, T.; Ryser, M.D. Growth Dynamics of Mammographic Calcifications: Differentiating Ductal Carcinoma in Situ from Benign Breast Disease. Radiology 2019, 292, 77-83, doi:10.1148/radiol.2019182599.
